# Differential and Cultivar-Dependent Antioxidant Response of Whole and Fresh-Cut Carrots of Different Root Colors to Postharvest UV-C Radiation

**DOI:** 10.3390/plants12061297

**Published:** 2023-03-13

**Authors:** Lucia Valerga, Roxana E. González, María B. Pérez, Analía Concellón, Pablo F. Cavagnaro

**Affiliations:** 1Consejo Nacional de Investigaciones Científicas y Técnicas (CONICET), Ciudad Autónoma de Buenos Aires CP1425, Argentina; 2Instituto Nacional de Tecnología Agropecuaria (INTA) Estación Experimental Agropecuaria La Consulta, La Consulta, San Carlos, Mendoza M5567, Argentina; 3Facultad de Ciencias Exactas y Naturales, Universidad Nacional de Cuyo, Mendoza M5502, Argentina; 4Centro de Investigación y Desarrollo en Criotecnología de Alimentos (CIDCA), La Plata B1900, Argentina; 5Facultad de Ciencias Exactas, Universidad Nacional de La Plata, La Plata B1900, Argentina; 6Facultad de Ciencias Agrarias, Universidad Nacional de Cuyo, Chacras de Coria M5528, Argentina

**Keywords:** UV-C radiation, fresh-cut processing, phenolic compounds, antioxidant capacity, anthocyanins, chlorogenic acid

## Abstract

Fresh-cut produce have become widely popular, increasing vegetable consumption in many parts of the word. However, they are more perishable than unprocessed fresh vegetables, requiring cold storage to preserve their quality and palatability. In addition to cold storage, UV radiation has been used experimentally to try to increase nutritional quality and postharvest shelf life, revealing increased antioxidant levels in some fruits and vegetables, including orange carrots. Carrot is one of the main whole and fresh-cut vegetables worldwide. In addition to orange carrots, other root color phenotypes (e.g., purple, yellow, red) are becoming increasingly popular in some markets. The effect of the UV radiation and cold storage has not been explored in these root phenotypes. This study investigated the effect of postharvest UV-C radiation in whole and fresh-cut (sliced and shredded) roots of two purple, one yellow, and one orange-rooted cultivar, with regard to changes in concentration of total phenolics (TP) and hydroxycinnamic acids (HA), chlorogenic acid (CGA), total and individual anthocyanins, antioxidant capacity (by DPPH and ABTS), and superficial color appearance, monitoring such changes during cold storage. Results revealed that the UV-C radiation, the fresh-cut processing, and the cold storage influenced the content of antioxidant compounds and activities to varying extents, depending on the carrot cultivar, the degree of processing, and the phytochemical compound analyzed. UV-C radiation increased antioxidant capacity up to 2.1, 3.8, 2.5-folds; TP up to 2.0, 2.2, and 2.1-folds; and CGA up to 3.2, 6.6, and 2.5-folds, relative to UV-C untreated controls, for orange, yellow, and purple carrots, respectively. Anthocyanin levels were not significantly modified by the UV-C in both purple carrots evaluated. A moderate increase in tissue browning was found in some fresh-cut processed UV-C treated samples of yellow and purple but not orange roots. These data suggest variable potential for increasing functional value by UV-C radiation in different carrot root colors.

## 1. Introduction

In recent years, a heightened awareness of the health benefits associated with a plant-based diet rich in antioxidant and anti-inflammatory compounds has led, among other factors, to an increase in the consumption of fruits and vegetables. Among the latter, fresh-cut vegetables represent convenient produce for consumers due to their practicality, readiness for consumption, and freshness [1]. Despite these advantages, these produce are generally highly perishable and present a short shelf-life as they are more susceptible to developing metabolic and physiological disorders, such as high respiration rates, excessive ethylene production, membrane deterioration, and occurrences of off-flavors, as well as exhibiting increased susceptibility to microbial development [2]. Ultimately, such changes can lead to reduced nutritional quality and palatability. Because these physiological and metabolic processes are intensified with higher temperatures, the use of low-temperature storage is currently the main postharvest technological tool for minimizing or delaying such defects [3].

In addition to cold storage, other technologies can be used to extend, and even increase, the postharvest quality of horticultural products, including fresh-cut fruits and vegetables. Among them, ultraviolet (UV) radiation has been applied as a germicide agent to control postharvest diseases and human pathogens, and as a stressor, for eliciting the biosynthesis and accumulation of nutraceutical compounds in several species [4]. Altogether, these studies have shown that both the germicidal effectiveness and the nature and extent of the phytochemical elicitation depend on technological variables, such as the wavelength, dose, and intensity of the UV radiation used; as well as on factors that are inherent to the product type, including the plant species, cultivar, organ maturity, and degree of processing (for fresh-cut products). Within the ultraviolet spectrum (100–400 nm), UV-B (280–315 nm) and UV-C (100–280 nm) radiations have been mostly utilized, with relatively fewer studies reporting on the use of UV-A (315–400 nm) [4]. UV-C, in particular, has been used for sanitizing purposes and for activating pathways associated with the production of health-enhancing compounds, mostly phenolics, in several whole and fresh-cut root, bulb, fruit, and leafy horticultural products.

Carrot is one of the most widely cultivated and consumed vegetables worldwide. Its tap root is predominantly consumed fresh, often in salads, with its previous fresh-cut processing being necessary. Although Western-type, orange-rooted carrots predominate in most parts of the world, there are other commercially-available cultivars that vary in root color (purple, red, yellow, and white), nutrient composition, and health-promoting properties. Such variation in root color and, to a substantial extent, nutritional value is due to the accumulation of different combinations of anthocyanin and carotenoid pigments, resulting in anthocyanin-rich purple or black carrots; orange carrots with high levels of β-carotene and α-carotene, both provitamin A carotenoids; lycopene-rich red carrots; yellow carrots that predominantly accumulate xanthophylls (lutein in particular); and white-rooted carrots, with nearly undetectable levels of the previous pigments [5,6]. Furthermore, varying combinations of these pigments, often localized in different root tissues (e.g., roots with anthocyanin-rich purple periderm and outer phloem, and carotene-rich orange inner phloem and xylem), occur among carrot cultivars. In addition to pigment variation, carrots of different color also vary in total phenolics concentration and phenolics composition, despite the fact that chlorogenic acid was the major non-anthocyanin phenolic compound found in most carrot cultivars analyzed to date [7,8]. In general, solid purple carrots have a greater concentration of total phenolics, non-anthocyanin phenolics, and chlorogenic acid, and greater antioxidant capacity than orange, red, yellow, and white carrots [6,7,8,9]. Variation in the carrot germplasm for concentration and composition of other phytochemical classes with health-enhancing properties has been reported for polyacetylenes [10] and terpenes [11]. Altogether, such genetic and compositional diversity has contributed to broadening the varietal and nutritional spectra offered to carrot consumers.

Although, generally, orange carrots have lower antioxidant capacity and total phenolics concentration than other vegetables [12], both of these variables could be increased in response to postharvest abiotic stresses, such as cutting the root to varying extents (e.g., in sticks, discs, quarters of discs, shreds, etc.), typically used in fresh-cut produce, and/or applying UV radiation [13,14,15,16]. In addition, comparison among fresh-cut processing (FCP) intensities in orange carrots revealed greater increases in total phenolic levels with greater FCP intensity (i.e., shreds > discs > slices) [14,16]. When both types of stresses (i.e., FCP and UV-C radiation) were applied combined to orange carrots, synergistic increases in total phenolic levels and antioxidant activity were found [17,18,19]. The effect of FCP and UV-C radiation has not been explored in carrots of other root colors.

In all of these previous studies, after the stresses were applied, the fresh-cuts and/or UV-treated carrots were stored at nearly ambient temperature conditions (15–20 °C), under which phenolic and antioxidant levels were monitored [17,18,19]. However, fresh-cut produce require refrigerated storage, using temperatures of 2–6 °C to help extend their quality and shelf life. Since the metabolic processes that are triggered in response to these stresses are temperature-dependent [14], results from these studies cannot be directly extrapolated to different storage temperatures. Thus, similar studies using UV radiation on fresh-cut vegetables, and monitoring antioxidant and compositional changes under cold-storage would provide a more realistic picture on the extent of the effects, and their time-course variation, that these postharvest practices induce in such fresh-cut produce. In orange carrots, only one study examined the effect of UV-C followed by storage at 5 °C, but the UV-C treatment was applied before fresh-cut processing (using shredded roots) [20]. Given that the effect of the UV radiation is rather superficial in a plant organ [i.e., it occurs in the outermost cell layers of the irradiated tissue [21]], and therefore dependent on the surface exposed to the UV, among other factors, the choice of applying the UV treatment before or after FPC can result in substantial differences with regard to increasing antioxidant levels, as the UV treatment is most effective when applied after FPC [22]. To date, the effect of UV radiation has only been tested in orange-rooted carrots. Given the broad differences in concentration and composition of pigments and other phytochemicals across carrots with different root color, and the increasingly growing market for these root phenotypes, as well as for fresh-cut vegetable products, it is of interest to examine their response to UV-radiation as a means to enhance their nutritional value in fresh-cut produce. Thus, the goal of the present study was to evaluate the response to UV-C radiation in whole and fresh-cut (sliced and shredded) roots from four carrot cultivars with different root colors (two purple, one yellow, and one orange) with regard to changes in phenolic and anthocyanin composition, antioxidant activity, and superficial color appearance, monitoring such changes during postharvest cold storage.

## 2. Results

### 2.1. Effect of UV-C Radiation on Carrot Antioxidant Capacity and Concentration of Total Phenolics, Hydroxycinnamic Acids, and Anthocyanins

Significant changes (*p* < 0.05) in the concentration of total phenolics (TP) and total hydroxycinnamic acids, estimated by the hydroxycinnamic index (HI), as well as in antioxidant capacity (AOX) (measured by the ABTS and DPPH methods), were found in response to the UV-C treatment in at least two of the three processing treatments in all of the carrot accessions examined (Figure 1, Figure 2, Figure 3 and Figure 4). In general, variations in TP and HI levels were concomitant with variations in AOX, with these traits being significantly and positively correlated (Table 1), suggesting that phenolic compounds account for a substantial fraction of the carrot antioxidant properties. Despite these general trends, the extent of such effects varied across accessions and among different sample types (i.e., whole, sliced, and shredded) within an accession. Below are detailed results for each carrot accession and root phenotype.

#### 2.1.1. Orange Carrot (cv. Maestro)

In whole roots of the orange-rooted cultivar Maestro, antioxidant capacity by DPPH and ABTS increased significantly after the UV-C treatment (relative to the untreated Control) since the sixth day of cold storage, after which AOX levels practically stabilized to finish—at the end of the storage (21 d)—with statistically comparable values (Figure 1A,D). Phenolics concentration in whole roots also increased in response to UV-C light, with significantly greater mean TP and HI values observed first at day 6 of storage, which remained statistically greater than their untreated controls throughout the end of the experiment (Figure 1G,J). For these samples, maximum increases of 1.8, 1.7, 2.0, and 1.3-folds, relative to their respective controls, were found for ABTS, DPPH, TP, and HI, respectively. Noteworthy, at day 21 of cold storage, none of the control samples had significantly greater mean values than their basal values at day 0 (prior to storage), suggesting that the cold storage did not influence phenolic and antioxidant levels.

For sliced roots, AOX, TP, and HI levels increased during storage in both control and UV-C treated samples, but the rate and extent of such increases were significantly greater in the UV-C treated carrots (Figure 1B,E,H,K). For all the variables, significant differences between the two treatments were observed from day 6 to the end of the experiment, with maximum differences of 2.1, 1.6, 2.0, and 1.8-folds found (at day 10 of storage) for ABTS, DPPH, TP, and HI, respectively.

For shredded carrots, AOX, TP, and HI levels increased during cold storage in both control and UV-C treated samples relative to their basal values (day 0), but no significant differences were found between the two treatments (Figure 1C,F,I,L).

#### 2.1.2. Yellow Carrot (cv. Uzbek Golden)

Whole roots of Uzbek Golden treated with UV-C exhibited significant increases in antioxidant capacity and in concentration of total phenolics and hydroxycinnamic acids, relative to the untreated controls, from day 6 to the end of the cold storage (day 21) (Figure 2A,D,G,J). Maximum differences between the two treatments were generally attained at the end of the cold-storage, with increases in the UV-C treated samples of 2.8, 2.4, 2.2, and 2.4-folds, relative to their respective controls, for ABTS, DPPH, TP, and HI, respectively. Noteworthy, all the control samples remained statistically invariable for all the variables throughout the experiment, suggesting that the cold storage did not influence phenolic and antioxidant levels.

For sliced roots, UV-C radiation was followed by steady and significant increases in antioxidant capacity, measured by both methods, and concentration of total phenolics and hydroxycinnamic acids, reaching maximums at the end of the storage (day 14); whereas, for the untreated controls, a moderate—yet significant—increase was observed for all the variables at day 6 of storage, which then decreased to values that were generally comparable to those of the basal status (day 0) (Figure 2B,E,H,K). For most of the variables, significant differences between the two treatments were observed from day 10 to the end of the experiment, except for DPPH, for which such differences were noted earlier (day 6) and continued until day 14. Overall, the UV-C treated samples exhibited maximum increases relative to their untreated controls of 2.6, 3.8, 2.2, and 3.8-folds for ABTS, DPPH, TP, and HI, respectively.

For shredded carrots, AOX, TP, and HI levels increased during cold storage in both control and UV-C treated samples relative to their basal values (day 0), but no significant differences were found between the two treatments for all the sampling times (Figure 2C,F,I,L). This suggests that only the fresh-cut processing, but not the UV-C treatment, influenced antioxidant levels in the shredded roots of Uzbek Golden.

#### 2.1.3. Purple Carrot (cv. Purplesnax)

In whole roots of Purplesnax, AOX, TP, and HI levels increased significantly during cold storage in both control and UV-C treated samples, reaching maximum values at day 6, followed by a moderate decline to the end at day 21, with levels that remained significantly greater than their respective basal values (day 0). The UV-C treated carrots had significantly greater levels of ABTS, DPPH, TP, and HI than the untreated controls for days 6, 12, and—for ABTS and HI—21 of cold storage (Figure 3A,D,G,J). Compared to their respective controls, the UVC-treated samples exhibited maximum increases in ABTS, DPPH, TP, and HI levels of 1.3, 1.5, 1.4, and 1.7-folds, respectively.

For sliced roots, antioxidant capacity (measured by both methods) and concentration of total phenolics and hydroxycinnamic acids increased significantly in the UV-C treated samples but not in their respective controls (Figure 3B,E,H,K). For ABTS and DPPH, significant differences between the two treatments were observed from day 10 to the end of the experiment, except for TP and HI, for which such differences were noted earlier (day 6) and continued until day 14. The UV-C treated samples exhibited maximum increases relative to their untreated controls of 1.8, 2.0, 2.1, and 2.0-folds for ABTS, DPPH, TP, and HI at the end of the cold storage (day 14), respectively.

A similar response to that of sliced roots was found for shredded roots, exhibiting increases in AOX, TP, and HI levels during cold storage in UV-C treated samples relative to both their basal values and respective controls, from day 10 to the end of the experiment (Figure 3C,F,I,L). UVC-treated samples showed maximum increases relative to untreated controls of 1.9, 1.7, 2.0, and 2.5-folds for ABTS, DPPH, TP, and HI, respectively.

Total anthocyanin contents did not vary significantly during the cold storage relative to their basal values, and the UV-C treatment did not influence anthocyanin levels in all the sample types (i.e., whole, sliced, and shredded roots) (data not presented).

#### 2.1.4. Purple Carrot (INTA44)

In whole root samples of INTA44, the time-course variation for AOX, TP, and HI levels was rather inconsistent and generally not significantly different across time-points and treatments (Figure 4A,D,G,J).

For sliced roots, AOX, TP, and HI levels increased during cold storage in both control and UV-C treated samples, but these increases were significantly greater in the UV-C treated carrots (Figure 4B,E,H,K). For DPPH and TP, significant differences between the two treatments were observed from day 6 to the end of the experiment, and for ABTS and HI since day 10. The maximum increases found in UV-C relative to control samples were 1.4, 2.5, 2.0, and 2.3-folds for ABTS, DPPH, TP, and HI, respectively.

In shredded roots, AOX, TP, and HI levels increased since day 6 in both UV-C and control samples, with these increases being significantly greater in the UV-C samples (Figure 4C,F,I,L). The UV-C treated carrots exhibited maximum increases relative to their respective controls of 1.8-folds for AOX and TP, and 2.0-folds for HI, between day 10 and 14. The total anthocyanin content in whole, sliced, and shredded roots did not vary significantly between the UV-C and control treatments during the cold storage (data not presented).

### 2.2. Dissecting the Effects of Fresh-Cut Processing and Cold Storage

In order to isolate and estimate the effects of FCP and cold storage from that of the UV-C radiation on phytochemical levels and AOX, whole vs. sliced vs. shredded root samples of UV-C untreated (control) carrots were compared before (at day 0) and after the cold storage (at day 21 and 14, for whole and sliced/shredded roots, respectively) (Figure 1, Figure 2, Figure 3 and Figure 4). In whole orange carrot roots, AOX, TP, and HI levels remained constant (statistically invariable) throughout the cold storage, whereas sliced and shredded roots exhibited significant increases for these variables at the end of the storage, suggesting that the FCP but not the cold storage induced an antioxidant response is the processed samples. However, it must be noted that an immediate decay in AOX, TP, and HI basal levels was observed due to the processing, prior to the cold treatment (day 0), with decreases of ~2-folds in sliced roots, and 2.5–4.3-folds in shredded roots. At the end of the cold storage, the levels of AOX, TP, and HI were statistically comparable between whole and fresh-cut processed roots.

A similar response to FCP was observed in the yellow carrot Uzbek Golden. For this cultivar, FCP prior to the cold treatment immediately reduced AOX, TP, and HI levels by ~2-folds in both sliced and shredded roots, following significant increases during storage for all of these variables. At the end of the storage period, whole and sliced root samples had comparable levels for all of the variables, whereas shredded roots exhibited significantly greater values than the formers. This suggests that in this yellow-rooted cultivar, the intensity of the FCP directly influences the antioxidant response.

In contrast to the effects observed in orange and yellow carrots, in the purple-rooted cultivar Purplesnax, FCP had no significant effects on AOX, TP, and HI levels, neither immediately after processing (day 0) nor at the end of the cold storage. In addition, whole roots of this cultivar revealed significant increases in ABTS, DPPH, TP, and HI levels at the end of the incubation period relative to their basal values and, for all of these variables except DPPH, whole roots had statistically higher levels than the fresh-cut samples at the end of the experiment. Total anthocyanin content was not affected by the cold storage or FCP. These data suggest that cold storage but not FCP induces the biosynthesis of phenolics, and perhaps other antioxidants, different from anthocyanins in Purplesnax. Finally, in the other purple-rooted cultivar, INTA44, FCP and cold storage had no significant effect on AOX, TP, HI, and anthocyanin levels.

### 2.3. Effect of UV-C Radiation on the Concentration of Chlorogenic Acid and Cyanidin Glycosides

In the present study, CGA was the main phenolic acid in all the carrot cultivars analyzed, with mean basal concentrations varying widely (~3-folds) among the different root color phenotypes and cultivars, as follows: 193.5 ± 34.5 mg kg^−1^ fw in the orange carrot Maestro, 299.7 ± 81.5 mg kg^−1^ fw in the yellow-rooted cultivar Uzbek Golden, and 366.8 ± 44.7 and 584.0 ± 55.9 mg kg^−1^ fw in purple carrots Purplesnax and INTA44, respectively. For all the cultivars, CGA levels in whole roots were significantly (*p* < 0.05) and moderate-to-strongly correlated with TP (r = 0.54–0.87), HI (r = 0.84–0.95), DPPH (r = 0.82–0.94), and ABTS levels (r = 0.80–0.87) (Table 1).

CGA contents in UV-C treated and untreated (control) sliced root samples, and in untreated whole roots of all the carrot accessions were estimated by means of HPLC analysis. Sliced root samples were selected as an intermediate intensity of FCP, and also because the major responses to the UV-C treatment were observed in this sample type, whereas untreated whole roots were used as negative controls of the FCP.

In the orange-rooted cultivar Maestro, CGA content in sliced roots increased significantly and steadily in response to the UV-C treatment relative to both controls (untreated sliced and whole roots) from day 6 to the end of the experiment (day 14), when maximum concentrations were reached, representing a 3.2-fold increase relative to the sliced root control (Figure 5A). CGA levels in both control treatments did not vary significantly from their respective levels at day 0, or between them during the cold storage.

As observed in orange carrots, in the yellow-rotted Uzbek Golden, CGA levels in sliced roots also increased significantly and steadily in response to the UV-C light exposure, as evidenced from day 6 to the end of the experiment (day 14) (Figure 5B). The sliced root control remained unaffected during the entire cold storage, whereas the whole root control showed a moderate yet significant increase at day 6 and then decreased to CGA levels, statistically similar to day 0. Overall, the rise in CGA content in the UV-C treated sliced samples represents a 6.6-fold increase relative to the content in the UV-C untreated controls.

As observed in the anthocyanin-free cultivars, CGA content in sliced purple roots of Purplesnax also increased significantly and steadily in response to the UV-C treatment, as evidenced from day 6 to the end of the experiment (Figure 5C). Both of the UV-C untreated controls showed little variation during the cold storage, ending with CGA mean values that were comparable to or below their initial content at day 0. Overall, the raise in CGA content in the UV-C treated sliced samples represents a 2.5-fold increase relative to the content in the control.

In the other purple-rooted carrot, INTA44, CGA concentration also increased significantly and steadily in response to the UV-C treatment, evidenced from day 6 to the end of the experiment (Figure 5D). The level of this phenolic in the sliced root control was unaffected during the cold storage, whereas the whole root control showed a moderate yet significant increase at day 6 and then decreased to CGA levels, statistically similar to day 0. Overall, the rise in CGA content in the UV-C treated sliced samples represents a 2.2-fold increase relative to the content in the respective control.

Anthocyanin HPLC analysis in root samples of Purplesnax and INTA44 identified the five major cyanidin glycosides reported previously for purple carrots [23]. The two cultivars exhibited substantial differences in their mean total anthocyanin content (465.0 vs. 30.1 mg kg^−1^ fw, for Purplesnax and INTA44, respectively), with profiles largely predominating in acylated pigments, accounting for 97.2% (in Purplesnax) and 95.7% (in INTA44) of the total anthocyanin content, although varying in the relative importance of their main compounds (Appendix A). In Purplesnax, the anthocyanin acylated with feruloyl (Cy3XFGG) largely predominated, accounting for 88% of the total pigment content; whereas, in INTA44, pigments acylated with sinapoyl (54.5%) and feruloyl (41.2%) predominated. None of these pigments were significantly influenced in their concentrations by the UV-C and/or the fresh-cut processing treatments, for both cultivars.

### 2.4. Quality Parameters

For whole roots, a good visual appearance was observed in both control and UV-C treated samples after 21 days of cold storage, comparable to their appearance prior to storage (at day 0) for all the cultivars (Figure 6). Only UV-C treated samples of yellow carrots presented some browning on the surface, as compared to control samples, but such an increase in tissue browning was not significant, as indicated by the statistically comparable browning indices in UV-C and control samples (Table 2). In addition, the weight loss in whole root samples was rather low (≤2.7%) (Table 2), suggesting a good postharvest quality after 21 days of storage for all the carrot accessions and root color phenotypes evaluated.

For sliced and shredded carrots, after 10 days of storage, they still exhibited an acceptable visual appearance, the main defect being a moderate browning of the tissues, most noticeable in the yellow-rooted samples (Figure 6). These samples reached weight losses of ~2–3% after 10 days of storage, nearly the same weight loss found in whole roots in twice the storage time (Table 2). Monitoring quality parameters after 10 days of storage revealed a progressive decline in these parameters after day 14, showing after this time point an incipient growth of microorganisms in both UV-C treated and control samples of sliced and shredded carrot of all root colors (data not presented).

In order to reflect the progress in tissue browning, the main deterioration symptom observed, a visual browning index assessment was conducted. In addition, the color change was objectively estimated by means of the ‘∆E’ parameter (described in Materials and Methods) (Table 2). In general, the browning index was higher in sliced and shredded roots after 10 days of storage than in whole roots after 21 days of storage. Similarly, mean ∆E values were higher in sliced and shredded carrots than in whole roots. Fresh-cut processed samples of yellow carrots were the most affected by tissue browning in response to the UV-C treatment, as indicated by, for example, the largest browning index (3.7) and ∆E values (24) found in sliced yellow roots.

## 3. Discussion

Fresh-cut vegetables are generally more perishable than their unprocessed counterparts, with the former requiring the use of postharvest technologies to increase their shelf-life. To date, the main postharvest technology used for this type of produce is cool storage, which slows down quality deterioration by reducing the rate of metabolic, physiologic, and phytopathogenic processes [24]. In addition to cold storage, other postharvest technologies, such as the use of UV radiation, have been shown to reduce the incidence of pathogen infections and even increase nutritional quality, mainly by stimulating phenolic biosynthesis in several vegetable species [4,25]. In fresh-cut orange carrots, UV-A, UV-B, and UV-C radiation have been previously used, revealing increases in total phenolics and chlorogenic acid levels, and antioxidant capacity [18,19]. However, these studies only examined orange-rooted materials and conserved the fresh-cut carrots under ambient temperature (15–20 °C), preventing the extrapolation of the results to cold storage conditions, which are necessary for preserving fresh-cut products. Thus, the present work complements previous studies, expanding the evaluation to other root color phenotypes of increasing relevance in vegetable markets worldwide, and using cold storage temperature (5 °C) to reflect typical conservation conditions used for fresh-cut products. Additionally, we estimated the effect of the fresh-cut processing and cold storage, independently from that of the UV-C treatment, for these novel carrot phenotypes.

The results from this study revealed that the UV-C radiation, the fresh-cut processing, and the cold storage influenced the content of antioxidant compounds and activities to varying extents, depending on the carrot cultivar, the degree of processing, and the phytochemical compound analyzed (Table 1; Figure 1, Figure 2, Figure 3 and Figure 4).

### 3.1. Effect of the UV-C Radiation

Regardless of the processing treatment, the UV-C radiation itself (i.e., comparing the UV-C treated carrots vs. their respective UV-C untreated controls for each processing treatment) significantly increased AOX (measured by both methods, ABTS and DPPH) in at least two of the three processing treatments, and in all cases the increase in AOX was paralleled with significant increases in phenolics concentration (TP and HI) (Figure 1, Figure 2, Figure 3 and Figure 4). However, marked differences were observed among the carrot cultivars and root colors. For instance, for the orange and yellow carrots, the UV-C radiation increased AOX and phenolic levels in whole and sliced roots (with greater increases found in the yellow carrot) but not in shredded roots (Figure 1 and Figure 2). Conversely, the purple-rotted cultivar Purplesnax exhibited significant increases in AOX and phenolic levels for all the processing treatments (Figure 3), whereas the other purple carrot, INTA44, increased AOX and phenolics in sliced and shredded but not in whole roots (Figure 4).

The little, or lack of, response to the UV-C radiation in shredded orange and yellow roots, as compared to their whole and sliced root counterparts (Figure 1 and Figure 2), has also been observed in a previous study by Surjadinata et al. [19], reporting that shredded orange roots had a much weaker UV-C response in terms of increases in AOX and total phenolics levels (as compared to UV-C untreated samples) than sliced root samples of the same cultivar. Moreover, Formica-Oliveira et al. [17] reported that shredded orange carrots irradiated with UV-C had comparable or slightly reduced total phenolic accumulation as compared to the UV-C untreated shreds during the incubation period, whereas Li et al. [26] reported lower phenolic accumulation in UV-C treated carrot sticks (5 cm × 0.5 cm × 0.5 cm) than in untreated sticks during conservation. Despite the fact that these studies used different UV-C doses and conservation temperatures than in the present work, Surjadinata et al. [19] used doses of 0.4, 0.7, and 10.6 kJ m^−2^; Formica-Oliveira et al. [17] used 4 and 9 kJ m^−2^; and Li et al. [26] used 2 kJ m^−2^; and all three studies incubated the samples at 15 °C, with comparable UV-C responses being observed for the most intense FCP treatment (i.e., shreds and sticks) in orange and yellow (in the present work) rooted carrots. Interestingly, Li et al. [26] found that concomitantly with their reduced phenolic accumulation, UV-C treated sticks also exhibited lower lignin content, and reduced enzyme activity for ‘phenylalanine ammonia-lyase’ (PAL), polyphenoloxidase (PPO), and peroxidase (POD), as well as lower expression of their encoding genes *DcPAL*, *DcPPO*, and *DcPOD* than their control counterparts, suggesting that the UV-C reduced or suppressed phenolic metabolism in these samples. Coincidently, Formica-Oliveira et al. [17] found reduced PAL activity associated with the lower phenolic accumulation in UV-C treated carrot shreds (compared to controls) and hypothesized that this could be due to partial PAL denaturation by the UV-C radiation. Whether the observed lack of response to the UV-C in the orange and yellow carrot shreds is due to the proposed partial denaturation of PAL and/or other enzymes of the phenylpropanoid pathway, or to the diversion of resources (e.g., sugars and amino acids) for the biosynthesis of cellular components other than phenolics—as described by Surjadinata et al. [19] as a means for coping with the combined stresses of the intense cutting and UV-C—is unknown and warrants further investigation.

In contrast to orange and yellow shreds, UV-C treated shreds of the two purple-rooted cultivars Purplesnax and INTA44 revealed increased TP, HI, ABTS, and DPPH levels, as compared to control shreds (Figure 3 and Figure 4). We hypothesize that the presence of anthocyanins in these samples may confer photoprotection against UV-C deleterious effects (e.g., denaturation of phenylpropanoid enzymes), thereby allowing the cells to elicit a more effective response in terms of phenolic biosynthesis, with a consequent rise in AOX. Photoprotection against ultraviolet light is one of the major roles proposed for anthocyanin pigments in plants [27]. From a fresh-cut produce consumer perspective, these unprecedented results with purple carrots suggest that (i) untreated purple shredded carrots are a healthier alternative to shredded orange carrots as they provide—on average—nearly 3-folds greater TP and AOX levels than the latter (estimates are for day 14 of conservation at 5 °C); (ii) UV-C radiation can be used to increase nutraceutical value in anthocyanin-rich carrots, as UV-C treated purple shredded carrots provided nearly twice the content of phenolics and AOX than their control counterparts; and (iii) the use of regular or UV-C treated purple carrot shreds can be used as an alternative to shredded beets in fresh-cut salads, alone or combined with orange shredded carrots, commonly sold in many markets. In addition to the reported effects in purple root shreds, the present study revealed, for the first time, significant and substantial increases in phenolics and AOX levels in sliced purple carrots irradiated with UV-C, with this response being consistent for the two purple-rooted cultivars analyzed (Figure 3 and Figure 4). Altogether, these data suggest that UV-C is an effective postharvest technology for further increasing antioxidant status in anthocyanin-rich carrots used for fresh-cut produce. Whether these results can be extrapolated to all purple carrots is unknown, and needs to be investigated in future studies including a larger number of genetically diverse purple carrot genotypes. Similarly, the significant increase in phenolics and AOX levels found in UV-C treated whole roots of Purplesnax (but not INTA44) suggests that this technology can be applied for increasing nutraceutical value in some, but not all, purple-rooted genotypes used for fresh consumption; although, the extent of its usefulness needs to be further evaluated in a broader purple carrot germplasm.

The comparison between the two purple carrots evaluated revealed differences in their response to the UV-C only for whole roots (the other processing treatments, sliced and shredded roots, yielded comparable results), with Purplesnax but not INTA44 showing significant increases in AOX and phenolic levels by the UV-C (Figure 3 and Figure 4). These discrepancies may be due to inherent differences in root morphology and size for these two cultivars, associated with the method used for preparing the extracts. The roots of Purplesnax are cylindrical and substantially thinner than the roots of INTA44, which have a conical shape and are thicker than the former (Appendix A); the mean ‘specific surface area’ (SSA) [calculated by the formula SSA = UV-C treated surface (cm^2^)/fresh weight of whole root (g)] was 0.72 for INTA44 and 0.91 for Purplesnax, representing ~26% difference in SSA. Thus, considering that the effect of the UV-C radiation in plants is rather superficial [i.e., it only affects the outermost cell layers of the irradiated tissue [22]], and that the extracts were prepared using the whole entire root, including also the inner root tissues that were unaffected by the UV-C, it becomes evident that the UV-C elicited tissues of INTA44 were combined with a larger proportion of unaffected (inner) root tissues for preparing the extracts, thereby “diluting” the UV-C response to a greater extent than in the whole roots of Purplesnax. Consequently, because of its lower SSA and increased dilution effect, the UV-C response in whole roots of INTA44 was, presumably, underestimated in the extracts, resulting in the absence of statistical difference relative to the control; whereas, in Purplesnax (with greater SSA), the superficial UV-C response was less diluted and therefore yielded significant increases in AOX and phenolic levels relative to its control. The UV-C treatment did not significantly increase total anthocyanin levels, estimated by spectrophotometry, nor the concentration of individual anthocyanin pigments, measured by HPLC, in the two purple rooted cultivars evaluated, for all the processing treatments. The fact that non-anthocyanin phenolics, such as hydroxycinnamic acids (Figure 3J–L and Figure 4K,L), and chlorogenic acid in particular (Figure 5C,D), were significantly upregulated in response to the UV-C radiation in these cultivars, but no increase was found for anthocyanins, suggests that the activation of the phenylpropanoid pathway by UV-C may occur after the synthesis of 4-Coumaroyl CoA (a common precursor to the synthesis of both chlorogenic acid and flavonoids) in the branch that leads to chlorogenic acid but not flavonoid synthesis (Appendix A). In partial agreement with this hypothesis, Bartley et al. [28] reported, in an orange carrot, transcriptional upregulation by UV-B of two of the three enzyme genes involved in the conversion of Coumaroyl CoA to CGA, namely ‘hydroxycinnamoyl CoAquinate hydroxycinnamoyl transferase’ (HQT) and ‘4-coumarate 3-hydroxylase’ (C3H), with their upregulation being correlated with significant increases in CGA content in the UV-B treated roots. However, they also reported upregulation of other genes earlier in the pathway, namely genes coding for PAL and ‘4-coumarate: CoA ligase’ (4CL), which agrees with other reports using UV-B in orange carrots finding increased contents of other phenolics generated earlier in the pathway, such as ferulic acid and isocoumarin [19]. Furthermore, Bartley et al. [28] also reported upregulation of one gene coding for chalcone synthase (CHS), the first enzyme in the flavonoids branch of the pathway. The upregulation of the latter does not agree with our findings for purple-rooted carrots, exhibiting no increase in anthocyanin level by UV-C. However, it must be noted that in the study of Bartley et al. [28], two CHS genes, termed *CHS1* and *CHS2*, were analyzed, and only *CHS2* was upregulated in response to UV-B. We performed alignment analysis of these gene sequences with the annotated carrot genome [29] to reveal that *CHS1* and *CHS2* correspond to carrot genes annotated as DCAR_030786 and DCAR_014462, respectively. Now, in the carrot genome, there are four CHS genes (with DCAR numbers 014462, 030785, 030786, and 000081) and only one of them (DCAR_030786) has been shown to participate in carrot anthocyanin biosynthesis in studies using transcriptome comparisons between purple and non-purple root tissues [30,31]. Thus, the gene found to be upregulated by UV-B in the study of Bartley et al. [28], namely *CHS2* (DCAR_014462), has not been shown to be involved in anthocyanin biosynthesis in purple carrots, whereas *CHS1* (DCAR_030786), known to participate in carrot anthocyanin biosynthesis, was not upregulated in their study. Thus, one can speculate that if their gene expression results using UV-B radiation could be extrapolated to purple carrots; the activation of this *CHS2* gene would not result in increased anthocyanin levels, as found in the present study. To date, this hypothesis has not been experimentally tested. In addition, the results from studies using UV-B and UV-C may not be directly comparable, as these light sources have different wavelengths and energies, and they may elicit biochemical responses by activating different genes of the pathway.

In the present study, CGA was the predominant phenolic acid in all the cultivars, with purple-rooted carrots being the ones with the greatest concentration of this compound, followed by yellow and orange-rooted carrots, with an overall range of variation of three-folds. In general agreement with our results, Sun et al. [6] analyzed CGA levels in seven carrot cultivars of different root colors, finding that CGA was the predominant phenolic acid in all the cultivars, with purple carrots exhibiting by far the greatest CGA concentrations, followed by red, white, orange, and yellow-rooted cultivars, reporting an overall range of variation of ~61-folds between the lowest and highest concentrations found. Altogether, the results from these two studies concur in the predominance of CGA over other phenolic acids, as well as in its relative content across carrots of different root color, but the range of variation found by Sun et al. [6] is substantially greater than found in this study. Such discrepancies may be due to differences in the carrot genotypes and number of materials analyzed, as well as differences in cultivation practices, environmental conditions, and analytical procedures between the two studies. In studies using only orange carrot cultivars, CGA was also found to be the main phenolic acid present in their roots [17,19]. In addition to its relative abundance in the root of all the cultivars, we found that CGA was the only phenolic compound significantly influenced by the UV-C radiation, as estimated by comparative analysis of the ‘area under the curve’ of different peaks in the HPLC chromatograms before and after the UV-C treatment (data not presented). In partial agreement with these results, previous studies using orange carrots have reported the greatest increases in CGA levels in response to postharvest UV-C radiation; although, smaller percentual increases were also noted in other phenolic acids and coumarins, namely ferulic acid, 4,5-dicaffeoylquinic acid (4,5-diCQA), isocoumarin, and isochlorogenic acid A (3,5-CQA) and C (4,5-CQA) [17,19]. As hypothesized earlier, this suggests that the UV-C light differentially activates branches of the phenylpropanoid pathway leading to the biosynthesis of phenolic acids, favoring, in particular, CGA production. In order to test this hypothesis, further studies that examine the full phenolic profile, ideally combined with comparative transcriptome analysis, of carrot roots in response to the UV-C radiation are needed.

### 3.2. Effect of the Fresh Cut Processing and Cold Storage

Time-course analysis of control samples of whole, sliced, and shredded carrot roots during the cold storage was used to dissect the effect of FCP from that of the UV-C treatment. As depicted in Figure 1, Figure 2, Figure 3 and Figure 4, FCP independently increased phenolic and AOX levels in three of the four carrot cultivars, as indicated by the significant increases observed in all the variables (TP, HI, ABTS, DPPH), except for anthocyanin content, in the fresh-cut (sliced and shredded) but not in whole root samples of Maestro, Uzbek Golden, and INTA44 (Figure 1, Figure 2 and Figure 4). In contrast, fresh-cut roots of Purplesnax did not follow the same trend, generally showing no statistical variation for all the variables during cold storage (Figure 3). According to Jacobo-Velázquez, González-Agüero [32], and Becerra-Moreno et al. [13], the FCP leads to a rapid increase in reactive oxygen species (ROS) production, which elicits a stress response, activating the phenolic metabolism to synthesize antioxidant compounds and lignin to counteract the oxidative stress and repair wounds. The end balance between synthesis and consumption of phenolic antioxidants (e.g., for lignin production or their degradation due to oxidation) in response to FCP may explain the different results observed among the cultivars, suggesting a much higher biosynthetic rate in Maestro, Uzbek Golden, and INTA44, allowing their accumulation in fresh-cut root tissues of these cultivars; whereas, in Purplesnax, such a balance seems to be rather neutral, not favoring one process over the other. However, it must be noted that in our analysis, the effect of the FCP cannot be separated from that of the cold storage, and it is possible that both factors are influencing phenolic accumulation in these samples.

In contrast to previous studies which incubated the samples at 15 °C after the UV-C treatment [17,18,19], the present work used 5 °C to simulate the cold storage conditions typically used for fresh-cut produce. Comparative analysis of non-irradiated (control) samples of whole, sliced, and shredded roots of the four carrot cultivars before and after the cold treatment, for all the variables analyzed, revealed varying effects due to the cold storage, depending on the carrot genotype, sample type, and variable analyzed (Figure 1, Figure 2, Figure 3 and Figure 4). In the orange rooted cultivar, both FCP samples (sliced and shredded) revealed significant increases in all the variables (TP, HI, ABTS, and DPPH) after 14 days of cold storage, with maximum increases observed in shreds, ranging from a 2.6 (for DPPH) to 5.9-fold increase (for TP), whereas whole roots were not significantly influenced by the cold storage except for DPPH which showed a small decrease (~25%) relative to basal values. A similar response was found in the yellow carrot Uzbek Golden, with significant increases revealed only for slices (for DPPH and TP) and shreds (for all the variables), with greater increases found in shreds, ranging from a 2.4 (for DPPH) to 3.4-fold increase (for HI). In the purple carrot Purplesnax, whole roots revealed significant increases during cold storage for all the variables analyzed, except for anthocyanin concentration, with increases in the range of 50% (for ABTS) to 70% (for DPPH), whereas the fresh-cut samples (slices and shreds) were not significantly influenced. In the other purple-rooted cultivar, INTA44, no significant effects due to cold storage were found for all the variables and sample types.

Although, in our study, no other conservation temperatures were evaluated, such as ambient temperature (used for other food products) or 15 °C (used in previous studies), and therefore we cannot speculate on how these variables under such conditions may compare to our actual data, this analysis estimated changes in antioxidant levels due to the cold storage for whole roots and different intensities of FCP. It appears that orange and yellow-rooted carrots had a very similar response to the cold storage, exhibiting greater increases in phenolics and AOX levels with greater FCP intensities, while little or no changes were observed in their whole root counterparts, suggesting that such changes were triggered by the cutting or wounding stress response, as reported in earlier studies with orange carrots [18,19,33]. Interestingly, the purple rooted cultivars were statistically unaffected by the cold storage for all the variables and sample types (in INTA44) or showed increased phenolic and AOX levels only for whole roots (in Purplesnax), contradicting the results found in orange and yellow carrots. While results in Purplesnax were unexpected, a closer examination of the data for INTA44 revealed substantially greater mean values for all the variables after the cold storage, but also showed large deviations from the means, which is typical for this highly-variable open pollinated cultivar (Cavagnaro, personal communication), presumably accounting for the lack of statistical significance for the observed increases.

### 3.3. Quality Parameters

Quality parameters of UV-C treated and control fresh-cut carrots after 10 days of cold storage exhibited very little weight loss (<3%) but revealed varying degrees of browning, depending on the cultivar, treatment, and sample type (Table 2 and Figure 6). While the UV-C treatment significantly increased browning in most of the fresh-cut samples of all root colors except orange, with sliced yellow carrots being the most affected by the UV-C, from a consumer point of view, such a degree of browning was still generally acceptable at day 10 of cold storage, but not at day 14 or later (data not shown). In order to reduce the degree of browning and increase consumer acceptance due to the visual appearance of these nutritionally superior fresh-cut produce, UV-C irradiated samples could be treated with typical anti-browning agents used in the food industry (e.g., ascorbic or citric acid) immediately after the light treatment, or irradiated samples could be stored under modified atmospheres to delay the browning [34]; although, whether such technological applications may interfere (or not) with the carrot UV-C response needs to be investigated.

Coincidently with our results, Alasalvar et al. [34] found moderate browning in shredded roots of purple but not orange carrots, although both root color phenotypes exhibited increased phenolic levels due to the cutting. According to Saltveit [35], when plant tissues are cut or wounded, two processes take place: (i) phenol oxidizing enzymes (e.g., PPO and POD) get in contact with their phenolic substrates and catalyze oxidation reactions of the latter, leading to tissue browning, and (ii) the phenolic metabolism is activated as a consequence of the stress-response in the wounded tissues, leading to the biosynthesis and accumulation of phenolic compounds. Thus, the fact that in our study, UV-C treated fresh-cut yellow and purple-rooted carrots presented some degree of tissue browning, but also high increases in phenolics concentration, suggests a greater phenolic biosynthetic rate, rather than consumption, for these compounds during cold storage. Such a balance between the rate of phenolic biosynthesis and degradation in UV-C treated and untreated fresh-cut carrots of different root color may also explain the different results found in orange vs. yellow and purple carrots. Presumably, the orange carrots used in the present study and in the study of Alasalvar et al. [34] could have a weaker oxidation response (perhaps due to a less active phenol-oxidation enzyme system) to the FCP than their purple and yellow-rooted counterparts. Whether these differences in the degree of tissue browning by the UV-C and/or FCP are genotype or root-color dependent needs to be further investigated.

## 4. Material and Methods

### 4.1. Plant Material, Cultivation Conditions, and Fresh Cut Processing

Two phenotypically different purple-rooted accessions were used: the commercial hybrid cultivar ‘Purplesnax’, obtained from Territorial Seed Company (Cottage Grove, OR, USA), with purple periderm and outer phloem, and orange inner phloem and xylem; and the open pollinated ‘INTA44’, developed by the carrot breeding program at INTA (Mendoza, Argentina), with purple periderm and orange phloem and xylem. In addition, seeds of the yellow-rooted open-pollinated cultivar ‘Uzbek golden’ were obtained from Baker Creek Heirloom Seeds, Mansfield, MO, USA, and seeds of the orange-rooted hybrid ‘Maestro’ were purchased from Vilmorin-Mikado, La Menitre, France. Appendix A presents the root phenotypes of the carrot materials used in this study.

The carrots were grown at the experimental field of the Faculty of Agricultural Sciences, National University of Cuyo (Mendoza, Argentina) in 2019, using conventional agricultural practices for the crop. Briefly, carrots were sown by hand and then, when the plants had 6–8 true leaves, they were thinned to ~80–90 plants/m^2^. The crop was drip-irrigated, and fertilized twice with Akaphos^®^ Violeta (Compo Expert Argentina, Ciudad de Buenos Aires, Argentina), containing N-P-K (13-40-13) and micronutrients, and adding a total of ~120, 360, and 120 kg per hectare of nitrogen (N), phosphorous (P_2_O_5_), and potassium (K_2_O), respectively. The field crop, from sowing to harvest, lasted 5 months, from 15 October 2019 to 15 March 2020. Appendix A present data for weather conditions and edaphic parameters at the carrot cultivation site.

When the roots reached commercial size [mean root length and thickness (at maximum diameter) were 19.7 and 3.4 cm for Maestro, 13.9 and 3.5 cm for Uzbek golden, 16.8 and 2.3 for Purplesnax, and 12.8 and 3.4 for INTA44], they were harvested, visually selected (only roots of similar size and without signs of pathogen attack or physiological disorders were kept), washed (to eliminate soil residues), dried, and processed. In order to standardize the sampling method and to minimize biases due to intra-root variation, the upper and bottom parts of the roots (~20% on each extreme) were removed, conserving only the middle section (~60% of the root). All the roots were used unpeeled. The roots (~200 per accession) were then subdivided into three lots of 60–70 roots each. One of the lots was used without further processing or cutting, henceforth referred to as ‘whole roots’, and the other two lots were processed by cutting to different extents; namely, root discs of 0.5 cm-thick, henceforth referred to as ‘sliced roots’, and in shreds of ~5.0 cm long × 0.4 cm wide × 0.4 cm thick, henceforth ‘shredded roots’. The carrots were fresh-cut processed with a slicer (Qsheng, Zibo, Shandong, China). Immediately after FCP, the samples were treated with UV-C, as described below. The control samples were not treated with UV-C.

### 4.2. UV-C Treatment, Storage Conditions, and Sampling

A cabinet (60 cm high, 40 cm deep, and 110 cm long) equipped with 4 UV-C light tubes (France, G30T8, 30 W, Shanghai, China) was used to produce UV-C radiation. Carrot root samples were placed inside rectangular plastic trays positioned 40 cm under the lamps and exposed to 1.36 mW cm^−2^ for 10 min (dose: 8 kJ m^−2^). Then, the root samples were turned over and treated again, using the same conditions and time, in order to expose both sides to the UV-C light. Whole and sliced samples were manually turned over, whereas the carrot shreds were turned over by flipping a planar surface onto which the shreds were evenly distributed prior to the UV-C radiation treatment.

The radiation intensity was determined employing a digital UV-C radiometer (Lutron UVC, shown to be involved in anthocyanin calculated as radiation intensity × treatment time). The UV-C conditions used were selected based on results from a previous pilot assay in which a range of radiation intensities and exposure times were evaluated on fresh-cut orange carrots (data not shown). After the UV-C treatment, samples of each processing treatment [each sample (or biological replicate) consisted of 30 g of fw for sliced and shredded roots, and 3 whole roots for the ‘whole root’ treatment] were placed in closed plastic boxes and stored in the dark at 5 °C and 91% of relative humidity for up to 14 or 21 days for fresh-cut (slices and shreds) and whole roots, respectively. During the cold storage, samples were taken after 0 (prior to cold storage), 3, 6, 12, and 21 days of storage, for whole roots; and after 0, 3, 6, 10, and 14 days of storage, for sliced and shredded roots. At each sampling time, three boxes of whole roots (i.e., nine whole roots in total), sliced roots, and shredded roots per treatment (i.e., UV-C and control) and accession were first visually examined and photographed, then analyzed in fresh for quality parameters (weight loss, browning index, and superficial color), followed by immediate sampling and storing at −80 °C for subsequent preparation of the extracts (one carrot extract was prepared from each sampling box) for antioxidant and compositional analysis. The carrot extracts were prepared as previously described [30]. Briefly, 2 g of the root tissues were mixed with 5 mL of methanol acidified with 10% formic acid, and the mixture was homogenized with a mortar and pestle, then transferred to a caramel glass bottle and incubated at 4 °C in the dark for 12 h, followed by centrifugation (11,000 rpm, 4 °C) for 15 min. Finally, the supernatant was separated and stored at −20 °C until analytical determinations were performed.

### 4.3. Quality Parameters

#### 4.3.1. Weight Loss

Each sampling box was weighted at the beginning (day 0) and throughout the storage period. Weight loss (WL) was calculated according to the formula: WL = (fw_i_ − fw_f_)/fw_i_ × 100, with fw_i_ and fw_f_ as the initial and final fresh weights, respectively. The results were expressed as a percentage.

#### 4.3.2. Visual Browning Index

A hedonic scale ranging from 1 (no browning) to 4 (severe browning) was used to visually estimate the extent of browning in the carrot samples throughout the storage period. A browning index value was assigned to each sampling box for each treatment and storage time.

#### 4.3.3. Superficial Color

The superficial color in the RGB scale was evaluated from photographs of the carrot samples taken under the same conditions using the software ImageJ [36], and then converted to Lab color space to finally calculate the ∆E parameter according to the formula:$$∆E=\sqrt{{\left(L_{2}^{*}-L_{1}^{*}\right)}^{2}+{\left(a_{2}^{*}-a_{1}^{*}\right)}^{2}+{\left(b_{2}^{*}-b_{1}^{*}\right)}^{2}}$$
where ∆E is the color difference between the start and end of storage (indicated with the subindex 1 and 2, respectively), and *L*, *a*, and *b*, are CIELAB color space. Three measurements of each image of the sampling box were performed for each treatment and storage time.

### 4.4. Determination of Total Phenolics (TP) Content

Samples (2 g) were macerated with 5 mL of extraction solvent (9:1 methanol: formic acid, *v*/*v*) and incubated overnight at 3 °C, without stirring. The obtained suspension was centrifuged at 3500× *g* for 15 min (Rolco, CM4080, Ciudad de Buenos Aires, Argentina) and the supernatant was recovered. Three independent extractions (one from each of three plastic boxes containing the carrot samples) were performed for each treatment, accession, and sampling time. Total phenolic content was determined using the Folin–Ciocalteu reagent, according to Singleton and Rossi [37] with minimal modifications. Briefly, 50 µL-aliquots of the extracts were mixed with 250 μL of Folin–Ciocalteu reagent, 1000 μL of Na_2_CO_3_ (20% *w*/*v*), and 3700 μL of distillated water. The absorbance at 765 nm was measured in a spectrophotometer (T60 UV/VIS, PG Instruments Ltd., Wibtoft, Lutterworth, UK) after 60 min of reaction. A commercial standard of chlorogenic acid (CGA) (Cayman Chemical, Buenos Aires, Argentina) was used as a reference to estimate TP levels, and results were expressed as CGA equivalents in mg kg^−1^ of fresh weight (fw). For each extract, three determinations were performed (i.e., three technical replicates).

### 4.5. Hydroxycinnamic Index (HI)

The hydroxycinnamic index was estimated according to Verardo et al. [38], with minor modifications. Briefly, 1 mL aliquots of the extracts, obtained as described in Section 4.4, were used to obtain absorbance readings at 330 nm in a spectrophotometer. A commercial reference compound of CGA was used as a standard to estimate the concentration of total hydroxycinnamic acids, expressed as CGA equivalents in mg kg^−1^ fw.

### 4.6. Total Anthocyanin Content

Total anthocyanin content was determined in the purple carrot samples of Purplesnax and INTA44 by the pH-differential method, according to Giusti and Wrolstad [39]. Briefly, two dilutions (1/100) of the extracts were performed, one with potassium chloride buffer (pH 1.0), and the other with sodium acetate buffer (pH 4.0). Absorbance readings at 530 and 700 nm for each dilution were obtained after 15 min of incubation. The total anthocyanin concentration in the original sample was calculated using the following equation:

Total anthocyanin content (mg L^−1^) = (A × MW × DF × 1000)/(ε × 1); where A is the absorbance, MW is the molecular weight, DF is the dilution factor, ε is the molar absorptivity, and 1 is the optical path length. A commercial standard of cyanidin-3-glucoside (Sigma-Aldrich, Inc., Atlanta, GA, USA) was used as a reference (MW = 449.2 g mol^−1^, ε = 34,300 L mol^−1^ cm^−1^) to estimate total anthocyanin concentration, expressed as mg of cyanidin-3-glucoside equivalents per kg^−1^ fw.

### 4.7. Chlorogenic Acid Determinations by HPLC Analysis

The carrot extracts were used to determine the concentration of CGA by means of high-performance liquid chromatography (HPLC) analysis, as described by Soto et al. [40] with very few modifications. Briefly, an ultra HPLC (UHPLC) apparatus (Shimadzu, SIL30, Chiyoda-ku, Tokyo, Japan) equipped with a binary pump system (LC-30AD, Nexera, Shimadzu), an autosampler (SIL-30ac, Nexera X2, Shimadzu, Chiyoda-ku, Tokyo, Japan), a photodiode array UV-VIS detector (SPD-M30A, Nexera X2, Shimadzu, Chiyoda-ku, Tokyo, Japan), and a C18 column (3 µm, 2.1 × 150 mm, UFLC Aqueous) was employed. The injection volume was 0.1 µL. Two mobile phases were used: water acidified with formic acid (0.1% *v*/*v*) as solvent A, and acetonitrile as solvent B. The gradient system was 0/5, 4/5, 17/30, 18/35, 19/50, 20/95, 21/95, 24/50, 27/5, and 37/5 min/% solvent B. Both solvents were filtered before use. Prior to injection, the samples were filtered through a 0.2 µm nylon syringe filter. CGA identification and quantification were based on retention times recorded at 330 nm, UV-Vis spectra, and standard curves constructed from using a commercial CGA reference compound (Sigma Aldrich, Atlanta, GA, USA). The results were expressed as mg kg^−1^ fw.

### 4.8. Anthocyanin HPLC Analysis

Anthocyanin profiles in the carrot extracts were obtained by HPLC analysis using the same equipment described above, following methods and conditions described by Perez et al. [41]. The mobile phase was distilled water acidified with 1% (*v*/*v*) formic acid as solvent A and methanol as solvent B. The gradient system was 0/5, 20/55, 21/100, 26/100, 27/5, and 40/5 (min/% solvent B). The injection volume was 0.5 µL. A commercial standard of cyanidin (Sigma Aldrich, Atlanta, GA, USA) was used for quantitation purposes and results were expressed as mg of cyanidin equivalents kg^−1^ fw.

### 4.9. Antioxidant Capacity

Antioxidant capacity in the carrot extracts was determined by means of the ABTS [2, 2-azino-bis (3-etilbenzotiazolin)-6-sulfonic acid] and DPPH (2, 2-diphenyl-1-picrylhydrazyl) methods, according to Arnao et al. [42] and Brand-Williams et al. [43], respectively. Briefly, for ABTS determinations, 50 µL of the carrot extract were added to 1 mL of the ABTS•+ working solution (absorbance at 734 nm was 0.700 ± 0.03) and the absorbance at 734 nm was measured after 6 min of reaction. For DPPH, 50 µL of the extract were mixed with 100 μL of a diluted solution of DPPH (40 mg L^−1^ in methanol; Abs ~1.1 at 515 nm). The absorbance at 515 nm was measured after 60 min of incubation at room temperature. Trolox^®^ (Cayman Chemical, Buenos Aires, Argentina) was used as a standard in both methods and results were expressed as Trolox equivalents antioxidant capacity (TEAC) in 5]^−1^ fw.

### 4.10. Statistical Analysis

Three biological replicates per treatment, accession, and sampling time were used in this study, and the results were expressed as mean values ± standard deviation (SD). The data were analyzed by ANOVA using the software InfoStat ver. 2009 [44] and means were compared using Fisher test, considering significant *p* values ≤ 0.05.

## 5. Conclusions

The present work investigated, for the first time, the effect of postharvest UV-C radiation in whole and fresh-cut carrots of different root colors for the concentration of different phenolic compounds (i.e., total phenolics, hydroxycinnamic acids, chlorogenic acid, and total and individual anthocyanins), antioxidant capacity (by DPPH and ABTS), and consumer quality parameters (dehydration, browning index, and color appearance) under cold storage conditions typically used for fresh-cut produce. Our data revealed that the UV-C treatment, the FCP, and the cold storage influenced phytochemical composition, nutritional value, and product quality to varying extents, depending mainly on the carrot genotype, root color, and intensity of the FCP. In general, significant and substantial increases were found for phenolics concentration, predominantly chlorogenic acid, and antioxidant capacity in all the cultivars evaluated, whereas no significant changes were observed for anthocyanin concentration and composition in the two purple-rooted cultivars evaluated. A moderate increase in tissue browning was found in some fresh-cut processed UV-C treated samples of yellow and purple but not orange carrots. Altogether, these data suggest that (i) the nature and intensity of the effects of postharvest UV-C radiation in carrot are genotype-dependent and strongly influenced by the degree of FCP; (ii) purple and yellow carrots are excellent alternatives for fresh-cut produce; (iii) the latter root phenotypes evidenced the greatest increases in phenolics and AOX levels in response to the UV-C treatment; and (iv) postharvest UV-C radiation can be effectively used to further increase nutraceutical value in anthocyanin-rich carrots by increasing the concentration of antioxidant phenolics other than anthocyanins.

## Figures and Tables

**Figure 1 plants-12-01297-f001:**
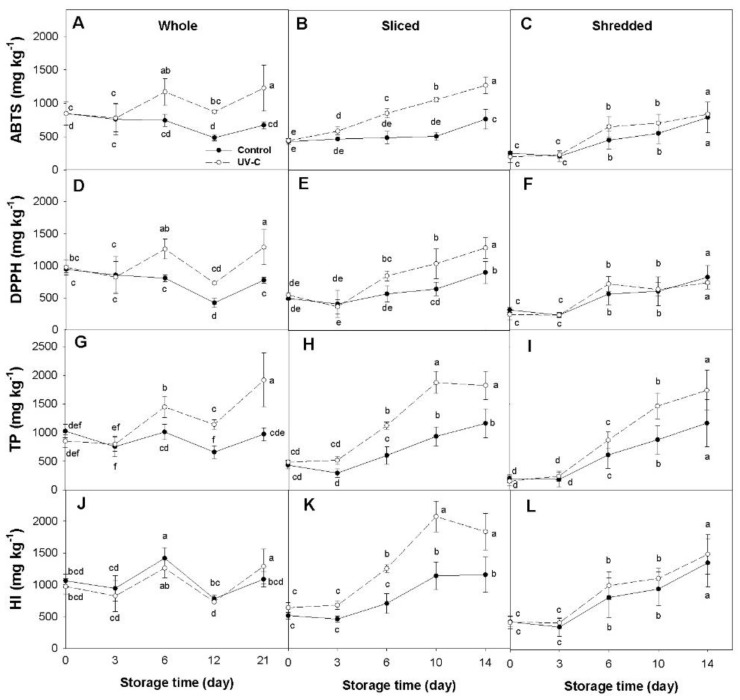
Time-course variation for mean antioxidant capacity determined by the ABTS (**A**–**C**) and DPPH (**D**–**F**) methods, concentration of total phenolics (TP) (**G**–**I**), and hydroxycinnamic index (HI) (**J**–**L**) values for whole (**A**,**D**,**G**,**J**), sliced (**B**,**E**,**H**,**K**), and shredded (**C**,**F**,**I**,**L**) root samples of the orange-rooted cultivar ‘Maestro’ during cold storage at 5 °C. Antioxidant capacity is expressed as mg Trolox equivalents per kg fw; and TP and HI as mg × kg^−1^ fw. Error bars represent standard deviation of the mean. Means not sharing a common letter are significantly different at *p* < 0.05, Fisher test.

**Figure 2 plants-12-01297-f002:**
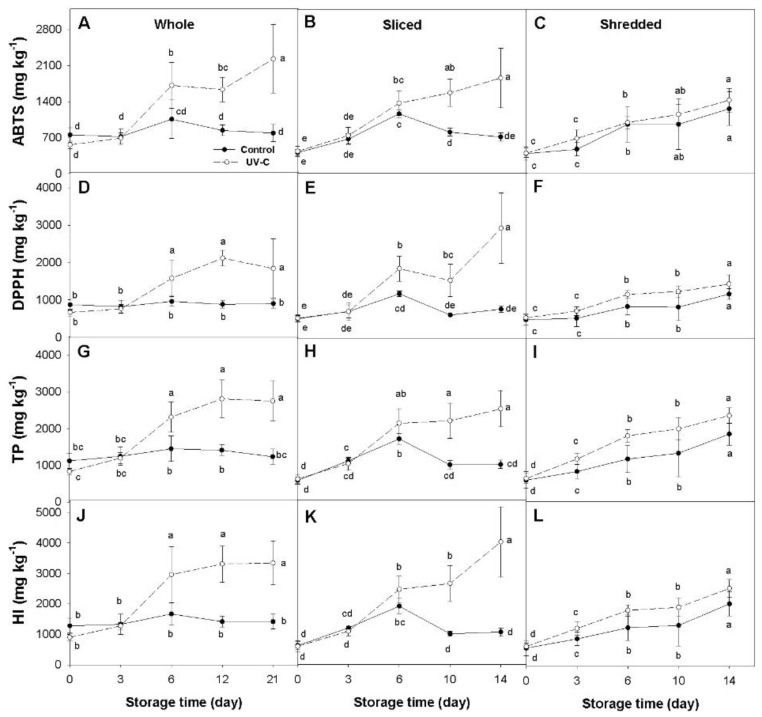
Time-course variation for mean antioxidant capacity determined by the ABTS (**A**–**C**) and DPPH (**D**–**F**) methods, concentration of total phenolics (TP) (**G**–**I**), and hydroxycinnamic index (HI) (**J**–**L**) values for whole (**A**,**D**,**G**,**J**), sliced (**B**,**E**,**H**,**K**), and shredded (**C**,**F**,**I**,**L**) root samples of the yellow-rooted cultivar ‘Uzbek Golden’ during cold storage at 5 °C. Antioxidant capacity is expressed as mg Trolox equivalents per kg fw; and TP and HI as mg × kg^−1^ fw. Error bars represent standard deviation of the mean. Means not sharing a common letter are significantly different at *p* < 0.05, Fisher test.

**Figure 3 plants-12-01297-f003:**
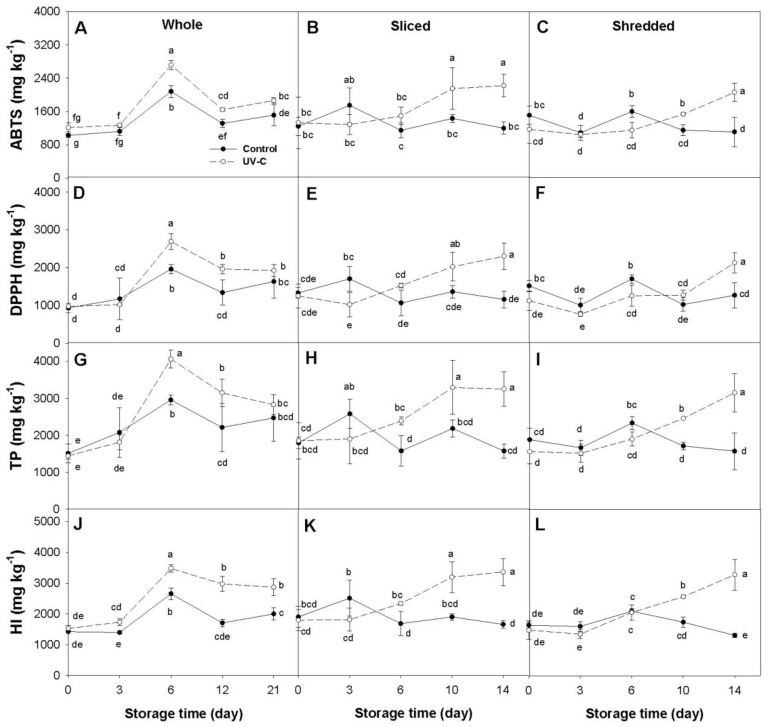
Time-course variation for mean antioxidant capacity determined by the ABTS (**A**–**C**) and DPPH (**D**–**F**) methods, concentration of total phenolics (TP) (**G**–**I**), and hydroxycinnamic index (HI) (**J**–**L**) for whole (**A**,**D**,**G**,**J**), sliced (**B**,**E**,**H**,**K**), and shredded (**C**,**F**,**I**,**L**) root samples of the purple-rooted cultivar ‘Purplesnax’ during cold storage at 5 °C. Antioxidant capacity is expressed as mg Trolox equivalents per kg fw, whereas TP, HI, and anthocyanin concentrations are expressed as mg × kg^−1^ fw. Error bars represent standard deviation of the mean. Means not sharing a common letter are significantly different at *p* < 0.05, Fisher test.

**Figure 4 plants-12-01297-f004:**
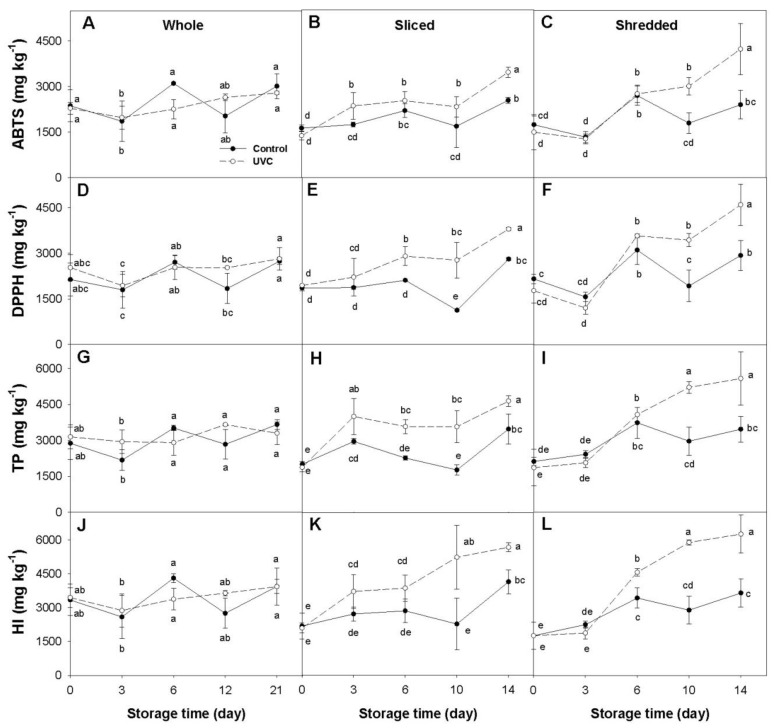
Time-course variation for mean antioxidant capacity determined by the ABTS (**A**–**C**) and DPPH (**D**–**F**) methods, concentration of total phenolics (TP) (**G**–**I**), and hydroxycinnamic index (HI) (**J**–**L**) for whole (**A**,**D**,**G**,**J**), sliced (**B**,**E**,**H**,**K**), and shredded (**C**,**F**,**I**,**L**) root samples of the purple-rooted inbred line ‘INTA44’ during cold storage at 5 °C. Antioxidant capacity is expressed as mg Trolox equivalents per kg fw, whereas TP, HI, and anthocyanin concentrations are expressed as mg × kg^−1^ fw. Error bars represent standard deviation of the mean. Means not sharing a common letter are significantly different at *p* < 0.05, Fisher test.

**Figure 5 plants-12-01297-f005:**
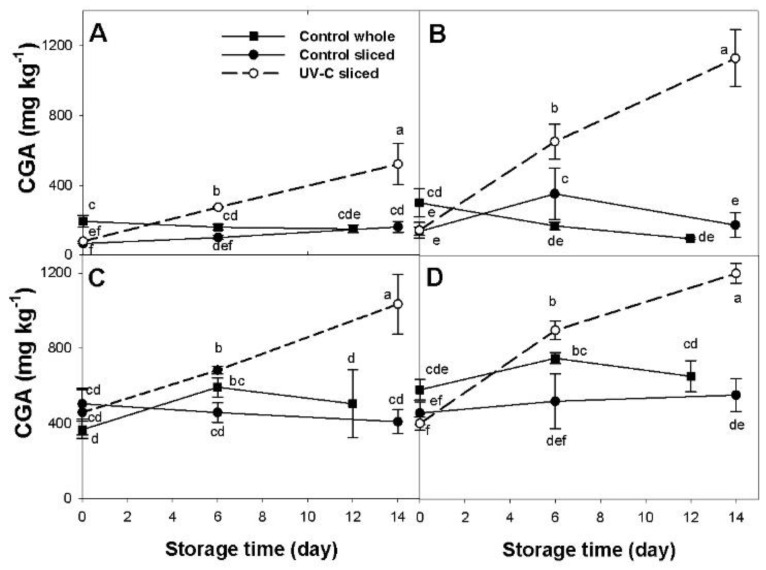
Time-course variation for chlorogenic acid (CGA) content in UV-C treated (- -○- -) and untreated (control) (-●-) sliced roots and in untreated whole roots (-■-) of the orange-rooted cultivar ‘Maestro’ (**A**), the yellow-rooted ‘Uzbek Golden’ (**B**), and the purple-rooted ‘Purplesnax’ (**C**) and ‘INTA44’ (**D**) during cold storage at 5 °C. Values are expressed as mg × kg^−1^ fw. Error bars represent standard deviation of the mean. Means not sharing a common letter are significantly different at *p* < 0.05, Fisher test.

**Figure 6 plants-12-01297-f006:**
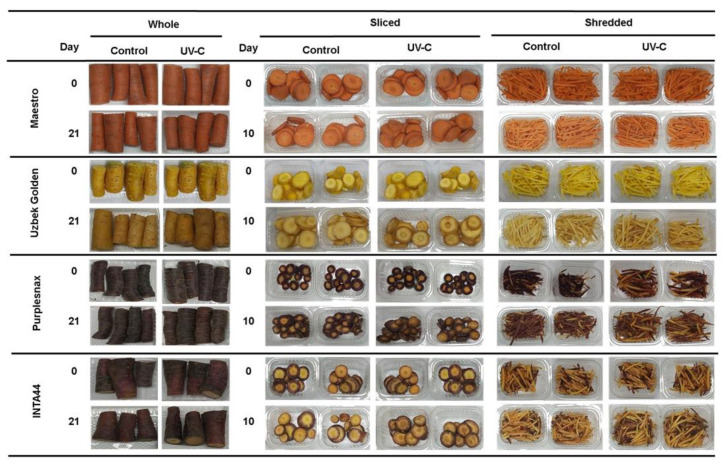
Appearance of UV-C treated and untreated (control) whole, sliced, and shredded root samples of orange (cv. Maestro), yellow (Uzbek Golden), and purple-rooted carrots (Purplesnax and INTA44), before the UVC treatment (day 0) and 10 (sliced roots) and 21 (whole roots) days after the UVC treatment and stored at 5 °C.

**Table 1 plants-12-01297-t001:** Pearson correlation coefficient (r) and *p* values among root concentration of different phenolics and antioxidant capacity estimates for four carrot cultivars with different root color *.

	Maestro (Orange)					Uzbek Golden (Yellow)					Purplesnax (Purple)					INTA44 (Purple)				
	ABTS	DPPH	TP	HI	CGA	ABTS	DPPH	TP	HI	CGA	ABTS	DPPH	TP	HI	CGA	ABTS	DPPH	TP	HI	CGA
ABTS	-	0.95	0.89	0.91	0.87	-	0.90	0.94	0.96	0.83	-	0.95	0.95	0.92	0.83	-	0.91	0.87	0.90	0.80
DPPH	<0.0001	-	0.83	0.87	0.85	<0.0001	-	0.89	0.97	0.94	<0.0001	-	0.95	0.91	0.91	<0.0001	-	0.89	0.89	0.82
TP	<0.0001	<0.0001	-	0.92	0.54	<0.0001	<0.0001	-	0.94	0.82	<0.0001	<0.0001	-	0.95	0.88	<0.0001	<0.0001	-	0.92	0.87
HI	<0.0001	<0.0001	<0.0001	-	0.88	<0.0001	<0.0001	<0.0001	-	0.92	<0.0001	<0.0001	<0.0001	-	0.95	<0.0001	<0.0001	<0.0001	-	0.84
CGA	<0.0001	0.0006	0.0003	0.0021	-	0.0011	<0.0001	0.002	0.0002	-	0.0048	0.0017	0.0040	0.0001	-	0.0079	0.0004	0.0017	0.0018	-

* Data are for phenolic contents and antioxidant capacities estimated with all the data combined (i.e., all treatments and storage time points). All the values were statistically significant at *p* < 0.05. ABTS: 2, 2-azino-bis (3-etilbenzotiazolin)-6-sulfonic acid; DPPH: 2, 2-diphenyl-1-picrylhydrazyl; TP: total phenolics; HI: hydroxycinnamic index; CGA: chlorogenic acid.

**Table 2 plants-12-01297-t002:** Percentage of weight loss, visual browning index, and color change (∆E) values in UV-C treated and untreated (Control) whole, sliced, and shredded root samples after 10 (for sliced and shredded roots) and 21 days (for whole roots) of cold storage, for orange (cv. Maestro), yellow (Uzbek Golden), and purple-rooted carrot accessions (Purplesnax and INTA44).

Carrot Accession (Root Color)		Whole		Sliced		Shredded	
		Control	UV-C	Control	UV-C	Control	UV-C
Maestro (orange)	Weight loss (%)	1.0 ± 0.3 ns	1.1 ± 0.2 ns	1.8 ± 0.5 ns	1.6 ± 0.2 ns	1.9 ± 0.9 ns	2.0 ± 02 ns
	Visual browning index *	1.0 ± 0.0 ns	1.0 ± 0.0 ns	1.3 ± 0.6 ns	2.0 ± 0.0 ns	1.3 ± 0.6 ns	1.3 ± 0.6 ns
	∆E ^§^	6.1 ± 0.9 ns	4.3 ± 2.3 ns	7.0 ± 4.6 ns	5.4 ± 3.0 ns	14.3 ± 3.0 ns	9.1 ± 1.4 ns
Uzbek Golden (yellow)	Weight loss (%)	1.1 ± 0.21 ns	1.5 ± 0.1 ns	1.5 ± 0.2 b	2.0 ± 0.2 a	1.7 ± 0.03 b	2.5 ± 0.1 a
	Visual browning index	1.9 ± 0.4 ns	2.1 ± 1.1 ns	1.7 ± 0.6 b	3.7 ± 0.6 a	2.7 ± 0.6 ns	3.7 ± 0.6 ns
	∆E	16.7 ± 2.1 ns	18.1 ± 1.6 ns	15.2 ± 2.6 b	23.9 ± 3.8 a	18.6 ± 1.3 ns	18.3 ± 2.4 ns
Purplesnax (purple)	Weight loss (%)	2.5 ± 0.2 ns	2.7 ± 0.01 ns	2.2 ± 1.3 ns	2.8 ± 0.2 ns	3.1 ± 1.2 ns	2.6 ± 0.3 ns
	Visual browning index	1.0 ± 0.0 ns	1.0 ± 0.5 ns	2.0 ± 0.0 b	3.0 ± 0.0 a	2.0 ± 0.0 b	3.0 ± 0.0 a
	∆E	4.7 ± 1.9 ns	4.3 ± 1.6 ns	3.4 ± 0.7 b	5.3 ± 0.7 a	8.3 ± 3.1 b	15.4 ± 1.2 a
INTA44 (purple)	Weight loss (%)	1.7 ± 0.6 ns	2.3 ± 0.7 ns	1.1 ± 0.1 b	1.8 ± 0.2 a	2.7 ± 0.5 ns	3.2 ± 0.3 ns
	Visual browning index	1.3 ± 0.5 ns	1.4 ± 0.5 ns	2.0 ± 0.0 b	3.3 ± 0.6 a	2.0 ± 0.0 b	3.0 ± 0.0 a
	∆E	9.4 ± 2.4 ns	10.7 ± 3.0 ns	10.6 ± 3.1 ns	5.9 ± 1.5 ns	13.1 ± 1.5 a	6.2 ± 2.2 b

* Values for the visual browning index ranged from 1 (no browning) to 4 (severe browning). ^§^ Color change was estimated by the formula $∆E=\sqrt{{\left(L_{2}^{*}-L_{1}^{*}\right)}^{2}+{\left(a_{2}^{*}-a_{1}^{*}\right)}^{2}+{\left(b_{2}^{*}-b_{1}^{*}\right)}^{2}}$, where ∆E is the color difference between day 0 (prior to storage) and end of storage (indicated with the subindex 1 and 2, respectively) and *L*, *a*, and *b*, are CIELAB color space. Values are means ± standard deviation. Different letters in the same line indicate significant differences between control and UV-C treatments for a given sample type at *p* < 0.05, Fisher Test. ns. not significant.

## Data Availability

All data are included in the main text and Appendix A.

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
