# Peer review of "Differential and Cultivar-Dependent Antioxidant Response of Whole and Fresh-Cut Carrots of Different Root Colors to Postharvest UV-C Radiation"

_plants, 2023, doi:10.3390/plants12061297_

Round 1

Reviewer 1 Report

The manuscript titled “Differential and cultivar-dependent antioxidant response of whole and fresh-cut carrots of different root colors to postharvest UV-C radiation” is about an up-to-date topic. Results of this research can be used in lots of countries. I think the authors examined a lot of factors of this research questions, therefore the paper is very detailed and useful for some other sectors. I have just some small comments to increase to quality of the paper.

Please add more detailed information about the growing conditions (e.g. length of growing period, climatic conditions – temperature, precipitation, sunny hours, type of soil, fertilization before or during the growing season, density during the growing).

It can be read in the line 707, that “when the roots reached the commercial size”. Please add more detailed information about the commercial size of the observed cultivars.

In the lines 713 and 714 it is written “The roots (~200 per accession) were then subdivided in three lots of similar sizes”. Please add the size categories or mark the exact size.

Reviewer 2 Report

Well-designed study and good results. It is a nice way of showing the difference and potential of new cultivars and colours available to the consumer and the grower. It is a good way to make those delicious and pretty-looking vegetables more popular. Thanks for that. 

I have placed comments within the PDF file - please refer to the notes there.

The only point of concern is the use of data in Figs 1-4 and table 2. I have commented there in the file as well. It seems it is the same data but shows different aspects. Ideally, the information could be presented in one version or with less detail in the table. It is rather confusing to the reader why the same date is shown again in such great detail.

Additionally, some of the discussion is more of a conclusion - also noted in the text - and may need to be either placed within the conclusion section or rewritten to be a discussion piece. 

I have added some notes to the M&M - as some aspects are not clearly described, e.g. how the samples were removed from the storage boxes. How much tissue was sampled? How often? And how was it treated? Stored as a whole in the freezer or directly extracted and stored for later analysis? Please add a section on sample taking and storage. This should be in such detail as that someone could repeat your study. 

Some of the references of formatting errors - I marked those I noticed but may have missed some as well 
